# Causally Focused Convolutional Networks Through Minimal Human Guidance

## Abstract

Convolutional Neural Networks (CNNs) are the state of the art in image classification mainly due to their ability to automatically extract features from the images and in turn, achieve accuracy higher than any method in history. However, the flip side is, they are correlational models which aggressively learn features that highly correlate with the labels. Such features may not be *causally* related to the labels as per human cognition. For example, in a subset of images, cows can be on grassland, but classifying an image as cow based on the presence of grassland is incorrect. To marginalize out the effect of all possible contextual features we need to gather a huge training dataset, which is not always possible. Moreover, this prohibits the model to justify the decision. This issue has some serious implications in certain domains such as medicine, where the amount of data can be limited but the model is expected to justify its decisions. In order to mitigate this issue, our proposal is to *focus* CNN to extract features that are *causal* from a human perspective. We propose a mechanism to accept *guidance* from humans in the form of activation masks to modify the learning process of CNN. The amount of additional guidance can be small and can be easily formed. Through detailed analysis, we show that this method not only improves the learning of causal features but also helps in learning efficiently with less data. We demonstrate the effectiveness of our method against multiple datasets using quantitative as well as qualitative results.

## 1 Introduction

Convolutional Neural Networks (CNNs) are more popular than any other techniques in image classification. The ability to automatically extract required features is one key factor behind the phenomenal success of these models. Image classification being used in critical application areas such as medicine, surveillance, and many others, CNNs could make a huge impact in these domains. However, when implementing artificial intelligence based systems in such domains, attributing the success of the application to accuracy alone is not sufficient. In such cases, these systems are expected to be justifiable as the decisions made by them may have huge impact on various factors with high risks.

Recently, it has been observed that CNNs are very much efficient to find correlation between features and labels and often extract features greedily following that principle (Shwartz-Ziv & Tishby, 2017; Tishby & Zaslavsky, 2015; Chaitin, 2015; Blier & Ollivier, 2018). In this process, often it may happen that these models learn correlations (Shen et al., 2017) which may not be justifiable from human perspective. In order to eliminate the effect of non causal correlations, CNNs need to be trained on huge datasets which may not be always possible in various domains like medicine. Thus learning the correct features efficiently from less data becomes an important problem in these areas.

Let us illustrate this using an example. Suppose we have a dataset with images of cows on grasslands and aeroplanes in blue-sky. It has been observed that grass is extracted as a feature for cow and sky is extracted as a feature for aeroplane (see Fig. 1). We have used Grad-Cam (Selvaraju et al., 2017) to generate the heatmaps to visualize the features extracted by the CNNs. These heatmaps reveal that the model is using irrelevant features for classification. Possible solutions to overcome this issue would be to add more images to the dataset or to re-balance it in order to remove the data bias. Contrarily, our objective is to utilize the available data efficiently and to make the models learn

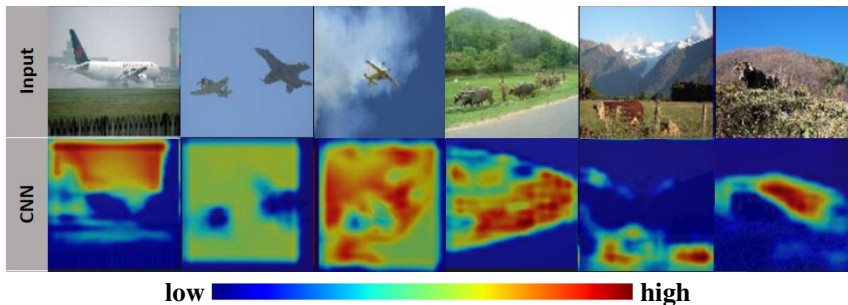

Figure 1: **CNNs need not learn causal features.** The examples are taken from the Pascal VOC 2012 dataset (first row) for aeroplane-cow classification in a biased dataset. In the second row we see the features learnt by the CNN highlighted using the Grad-Cam method. It is evident that the model has learnt features such as the sky and grassland for classifying aeroplane and cow respectively.

features which can be *causal* from human perspective. It is evident that, CNNs are not guaranteed to extract such *causal features*. With this point of view, we propose to take guidance from humans on what they think is causal for a few samples in a class. We capture this guidance in the form of activation masks which are basically binary matrices with 1s on the causal parts of the images (see Fig. 2). Once we have the user guidance, our plan is to tweak the learning process of the CNNs though these guidance to *focus* them on extracting the causal features. We achieve that by modifying the learning objective of the CNNs and the backpropagation algorithm then takes care of updating the model parameters accordingly. This simple modification in the training procedure helps avoid the learning of spurious correlations between features and labels and focus just on the causal ones. We have experimentally observed that, this concept is working quite well on a wide range of cases and has proved to be extremely useful in the case of medical datasets.

The main contributions of our work are summarized below:

1. We propose a technique to focus the CNNs in learning causal features with the help of minimal human guidance.

2. We demonstrate that our method not just improves learning of causal features but also helps in learning efficiently with less data. Additionally, we also show that the features learnt using our method are more robust to various types of image perturbations.

## 2 RELATED WORK

**CNN and Interpretability** Convolutional Neural Networks (LeCun et al., 1999) have boosted the progress in the field of computer vision since their inception. Manually designed architectures like LeNet (LeCun et al., 1989), AlexNet (Krizhevsky et al., 2012), VGG16 (Simonyan & Zisserman, 2014) and many more have been proposed in the literature. In order to simplify the CNN architectures by retaining the spatial structure throughout the network, Springenberg et al. (2014) proposed the all convolutional nets, which eliminate the fully connected layers in these networks. To interpret the decisions of the CNNs, tools like Grad-Cam (Selvaraju et al., 2017) are widely used in practice, as they provide a way to extract the class discriminatory features learnt by the model.

**Correlation and Causality** Research on the topics of correlation and causality has been gaining popularity among the researches in the recent years. Work by Shen et al. (2018) has recently shed some light on the correlational behavior of CNNs in image classification. Few other works like Arjovsky et al. (2019), try to understand the causal relation between the input images and the corresponding labels, i.e. studying whether the relation is causal, anticausal or agnostic in nature. In our work, we just rely on the fact that there exist few features in the images which are the cause for the label and we expect the model to correctly identify such causal features only. The importance of causality, specifically in field of medicine is studied by Castro et al. (2020) and Liu et al. (2019), highlighting the challenges for causality in computer aided diagnosis.

Table 1: This table summarizes a list of all the notations used in the paper.

| Symbol | Description | Symbol | Description |
|---|---|---|---|
| $X$ | Input image | $C$ | Number of classes |
| $y$ | One-hot class label | $F$ | Number of feature map outputs |
| $A$ | Input activation mask | $c$ | Index of true class |
| $\hat{y}$ | Predicted class probabilities | $k_1 \times k_2$ | Size of the filter |
| $\{\hat{A}_f\}_{f=1}^{F}$ | Set of feature map outputs | $m_1 \times m_2$ | Size of the input image |
| $W$ | A single filter in the convolution layer | $n_1 \times n_2$ | Size of the activation masks and the feature maps |
| $L_{cl}$ | Classification loss | $\alpha$ | Weight for the causally-focus loss |
| $L_{cf}$ | Causally-focus loss | $\epsilon$ | A very small constant value |

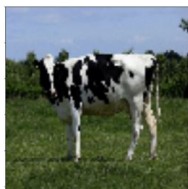 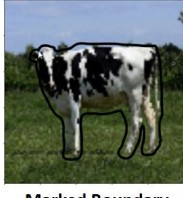 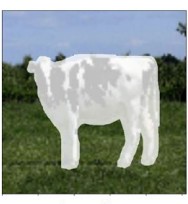 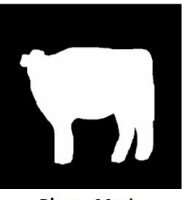

**Input Image**      **Marked Boundary**      **Selected Area**      **Binary Mask**

Figure 2: **Activation mask creation:** In the case of lack of annotated data, one can create activation mask for guidance very easily (this can be implemented in Python). The order is from left to right.

**Learning Causal Features** The very recent work by Xiao et al. (2020), studies the influence of the image background on object recognition. They show that non-trivial accuracy can be achieved by relying just on the background features in the images. A similar study was done in the case of medical images by Maguolo & Nanni (2021), where they showed that CNN models provided diagnosis for the chest x-ray images even when the lung regions were removed from the input images. Not much work has been done on improving the learning of causal features, especially in the case of small datasets. The closest work that we found to our method is the self-supervised method called Guided Attention Inference Network (GAIN) (Li et al., 2019) which is proposed to improve the priors for the task of weakly supervised image segmentation. The authors present an extended version of this method, called the $GAIN_{ext}^{p}$, which uses an additional parameter sharing network with the GAIN architecture for pixel level supervision, that brings up the similarity with our work. We use this model as a baseline in our experiments.

## 3 PROPOSED METHODOLOGY: CAUSALLY FOCUSED CONVOLUTIONAL NETWORKS (CFCN)

In this section we describe the proposed method dubbed, Causally Focused Convolutional Networks (CFCN). In CFCN, we force the model to break the spurious correlation between the label and any feature, and *focus* only on the *causal* features. In order to achieve this, we resort to minimal human guidance through activation masks.

### 3.1 NOTATIONS

Consider an input image $X$ of size $m_1 \times m_2$, its ground truth one-hot label $y$ and the corresponding input activation mask $A$. Let, $C$ be the number of classes in the dataset. The image classification model outputs class probabilities $\hat{y}$ and the set of feature maps $\{\hat{A}_f\}_{f=1}^{F}$ generated by the last convolutional layer after application of relu activation, where $F$ is the number of filters in this convolutional layer. The input activation masks and the feature map outputs are resized to a common shape $n_1 \times n_2$. Let $c$ denote the index of the true class of the input image. $A \circ B$ denotes the element-wise product of two matrices $A$ and $B$ of same size. All the notations are summarized in Tab. 1.

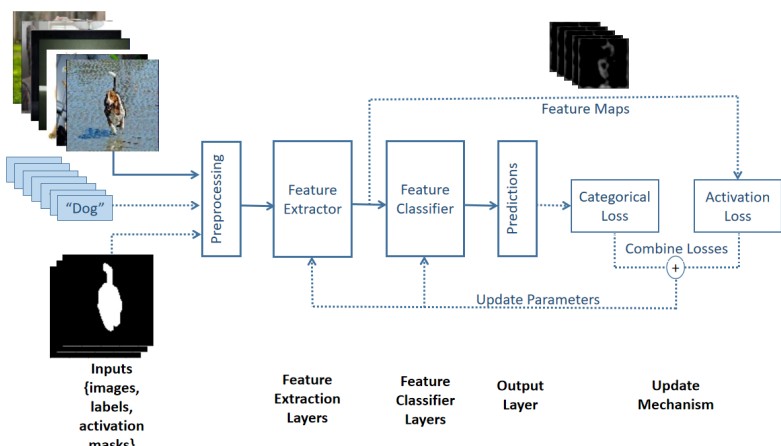

Figure 3: **Illustration of the methodology:** Along with images and labels, activation masks are also input for guidance. Dotted lines denote the flow present only during the training phase.

## 3.2 ACTIVATION MASK FOR HUMAN GUIDANCE

Activation masks are binary images where the causal regions are indicated using 1s and those belonging to the context regions are indicated using 0s. Some datasets like the Brain MRI (Cheng et al., 2015), readily provide binary masks which can be directly used for our purpose. In few other cases, we may have pixel level labels which provide fine grained annotations exactly covering the regions of the objects of interest or bounding boxes annotations, that provide relatively coarse regions which may also contain few context features in them. Such annotations can be used to generate the activation masks as described in Appendix A.

However, the method should not be dependent on the availability of masks or not. So we devise a simple technique to generate masks automatically. A typical step by step procedure for activation mask generation is shown in Fig. 2. For a small subset of training images, the user has to roughly select the area of the objects of interest, which is then converted into a binary mask as shown in the figure. This can be automated using a python script.

## 3.3 CAUSAL FOCUS THROUGH ACTIVATION MASKS

In general CNNs are composed of several convolutional layers and pooling layers to extract features. The features extracted at the last layer then get passed to a Feed-forward Neural Network (FNN) to assign labels. Better the quality of features better will be the performance of the classifier. CNN layers along with the FNN layers are trained end to end by optimizing the categorical cross entropy loss.

It has been observed that CNNs learn to extract features greedily and often end optimizing the correlation between labels and features. This process does not ensure that these models will always extract features which are causal from a human point of view. To mitigate this issue, we propose a mechanism to guide CNNs to focus on causal (in a human eye) features through additional minimal human input in the form of activation masks.

During the model training we provide input images, their labels and activation masks. For the subset of input images which do not have the activation masks, we provide dummy masks with all values as 1s. The forward pass through the network for a single input image $X$, with true label $y$ and input activation mask $A$ generates the class probabilities $\hat{y}$ and the feature map outputs $\{\hat{A}_f\}_{f=1}^{F}$ from the last convolutional layer, where $F$ is the number of filters in this layer. Using this notation, we

Table 2: Brief summary of the datasets used in the experiments.

| Dataset | Size | Significance |
|---|---|---|
| *Oxford IIIT Pets* | 7349 | Large annotated dataset. |
| *Aeroplane-Cow* | 718 | Small and biased dataset created from 'aeroplane' and 'cow' classes in Pascal VOC 2012 dataset (Everingham et al.). |
| *Brain MRI* | 3064 | Small medical dataset. |
| *IDRiD* | 82 | Small medical dataset. |

propose to optimize the following loss to train the CNN.

$$\mathcal{L} = \underbrace{-\sum_{i=1}^{C} y_i log(\hat{y}_i)}_{L_{cl}} + \underbrace{\alpha \left( 1 - \frac{1}{F} \sum_{f=1}^{F} \left( \frac{\sum_{j=1}^{n_1} \sum_{k=1}^{n_2} (A \circ \hat{A}_f)_{j,k}}{\sum_{j=1}^{n_1} \sum_{k=1}^{n_2} (\hat{A}_f)_{j,k} + \epsilon} \right) \right)}_{L_{cf}}, \tag{1}$$

where $\epsilon \geq 0$ is a small quantitiy to avoid accidental divide by zero error. $\alpha \geq 0$ is the trade-off parameter between the traditional categorical cross entropy loss ($L_{cl}$) and the proposed *causally-focus loss* ($L_{cf}$). Greater the value of $\alpha$ greater is the weightage on the causally-focus loss.

Apart from the traditional CNNs, we also applied our causal feature learning method to the all convolutional nets proposed by Springenberg et al. (2014). The number of filters in the last convolutional layer of these nets, is equal to the number of classes with each feature map output corresponding to each class, thus highlighting only that class specific features. We then calculate the causally-focus loss only with respect to the feature map $A_c$ corresponding to the true class of the image and the input activation mask $A$ as follows:

$$\mathcal{L} = \underbrace{-\sum_{i=1}^{C} y_i log(\hat{y}_i)}_{L_{cl}} + \underbrace{\alpha \left( 1 - \left( \frac{\sum_{j=1}^{n_1} \sum_{k=1}^{n_2} (A \circ \hat{A}_c)_{j,k}}{\sum_{j=1}^{n_1} \sum_{k=1}^{n_2} (\hat{A}_c)_{j,k} + \epsilon} \right) \right)}_{L_{cf}}, \tag{2}$$

where $c$ is the index corresponding to the actual class of the image, i.e. $y_c = 1$. This formulation has the ability to preserve the spatial structure of the data which is otherwise not maintained by the fully connected layers. Secondly, in Eq. 1, we are calculating the causally-focus loss on all the feature map outputs which can be more time consuming. A detailed analysis of the loss function is presented in the Appendix B. The proposed approach CFCN is depicted in Fig. 3.

## 4 EXPERIMENTS

### 4.1 DATASET

We have particularly selected the following four datasets - *Oxford IIIT Pets* (Parkhi et al., 2012), *Aeroplane-Cow*, *Brain MRI* (Cheng et al., 2015) and *IDRiD* (Porwal et al., 2018) - for performance comparison with the baseline. These datasets, help us in demonstrating the effectiveness of our method across different challenges such as biased data, small dataset size and feature extraction in medical images. The detailed description of these datasets is given in Tab. 2.

### 4.2 BASELINES

In our experiments, we use the CNN traditionally trained using just the classification loss as our first baseline. As another related method that uses additional pixel level guidance, we use the extended version of GAIN (Li et al., 2019) as our second baseline. In comparison to these, we compare two of our models: one that uses fully connected layers for classification (CFCN-F) and another one that uses the all convolutional nets (CFCN-C). We use the same architecture for feature extraction in all the methods for a given dataset.

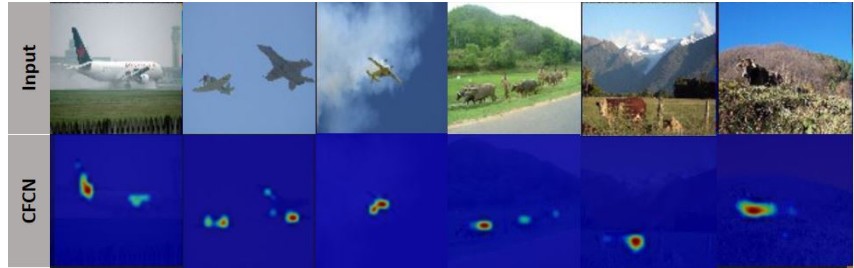

Figure 4: **Demonstrating the effectiveness of proposed CFCN:** This figure shows the counterpart heatmaps from CFCN for the images in Fig. 1. We see that unlike CNN, CFCN is able to focus on the causal features belonging to the aeroplanes and the cows present in the images.

Table 3: Quantitative results comparing proposed CFCN variants with the state of the art. We see that in the case of small datasets, CFCN models outperform the baselines in almost every metric. In the case of *Oxford IIIT Pets* dataset, CNN is performing slightly better than other methods but it must be noted that high accuracy may also come from the spurious correlations in the features and the labels. This fact is also evident as shown in the visual results (Fig. 5).

| Models | Metrics | | | Metrics | | |
|---|---|---|---|---|---|---|
| | Acc. | F1 | ROC | Acc. | F1 | ROC |
| | *Oxford IIIT Pets* | | | *Aeroplane-Cow (Small Dataset)* | | |
| **CNN** | **0.94** | **0.94** | **0.98** | 0.78 | 0.74 | 0.83 |
| **GAIN** | 0.86 | 0.84 | 0.92 | 0.72 | 0.70 | 0.74 |
| **CFCN-F** | 0.86 | 0.85 | 0.97 | 0.76 | 0.75 | 0.83 |
| **CFCN-C** | 0.90 | 0.88 | 0.96 | **0.82** | **0.81** | **0.86** |
| | *Brain MRI (Small Dataset)* | | | *IDRiD (Small Dataset)* | | |
| **CNN** | 0.86 | 0.85 | **0.97** | 0.83 | 0.80 | 0.24 |
| **GAIN** | 0.66 | 0.56 | 0.83 | 0.80 | 0.84 | **0.82** |
| **CFCN-F** | **0.88** | **0.87** | 0.96 | **0.92** | **0.94** | 0.80 |
| **CFCN-C** | **0.88** | **0.87** | **0.97** | 0.88 | 0.90 | 0.71 |

## 4.3 Experimental Setting

All our experiments were run on a GPU system with 16 GB RAM and a single GeForce RTX 2080 GPU. The codes are implemented in Python 3.7 with Tensorflow v2.2. We have used the 'matplotlib' library in Python to generate all the plots and used the 'polyfit' function available in 'numpy' library to regress a curve in the plots wherever necessary. As a preprocessing step, we normalize the input images in the range [0,1]. We evaluate the performance of all the models using five quantitative metrics: accuracy, macro f1-score and AU-ROC. Further we present the qualitative results using Grad-Cam for heatmap visualization. A detailed description of the experimental setup for each dataset is given in the Appendix C

## 4.4 Results: Beyond CNNs to Detect Causal Features

Recall from Fig. 1 that CNNs need not learn causal features as it optimizes over correlation. Whereas, with additional human guidance, CFCNs are able to ignore the spurious correlations present in the data and learn just the causal features in the images as shown in Fig. 4.

Tab. 3 compares the performance of various models on several metrics. It can be seen that both CFCN variants perform quiet well. Furthermore, we see that, in the case of very small datasets (*Aeroplane-Cow, Brain MRI and IDRiD*), our models outperform the baselines in almost every metric. Further, to validate the correctness of our models in terms of feature learning, we present the comparison of the Grad-Cam heatmaps generated by these models in Fig. 5. We observe that CFCNs

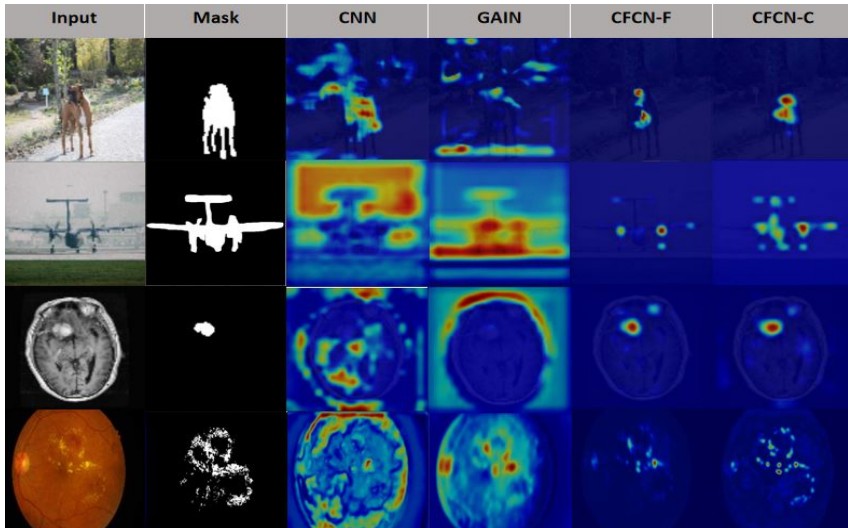

Figure 5: **Comparison of models on qualitative results:** CFCN variants learn causal features whereas the features learnt by baselines are not causal.

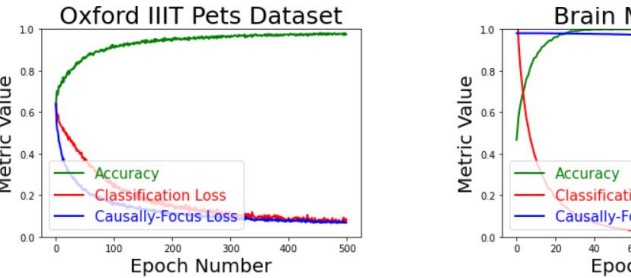

Figure 6: **Convergence:** Causally-focus loss does not affect the convergence in training CFCNs.

are not just accurate but also rely on the class-discriminatory causal features in the images for classification rather than using any context information. We share a few additional results in the Appendix D.

**The Convergence of CFCN.** We also empirically verify that adding the causally-focus loss does not affect the training of the underlying CNN. Fig. 6 shows steady convergence based on both the loss functions.

## 4.5 RESULTS: EFFECT OF VARYING THE WEIGHTAGE FOR CAUSALLY-FOCUS LOSS

As we have seen that, CFCN is able to detect causal features and also perform better or quite similar to the baselines, we want to investigate the sensitivity of the setup of human guidance.

The trade-off parameter $\alpha$ is an important quantity as it trades-off between the classification loss and causally-focus loss (see Eq. 1 and Eq. 2). We expect that as the value of $\alpha$ increases, the model should focus more and more on the causal features. We see the expected behaviour in Fig. 7. Further, we have observed that beyond a certain threshold, the activations become very small there-by affecting the classification performance of the models.

## 4.6 RESULTS: LEARN MORE WITH LESS DATA DUE TO CAUSALITY

Through this experiment we demonstrate that our method helps not just in learning causal features but also helps in learning faster with less data. The basic idea is, with more guidance, we reduce the

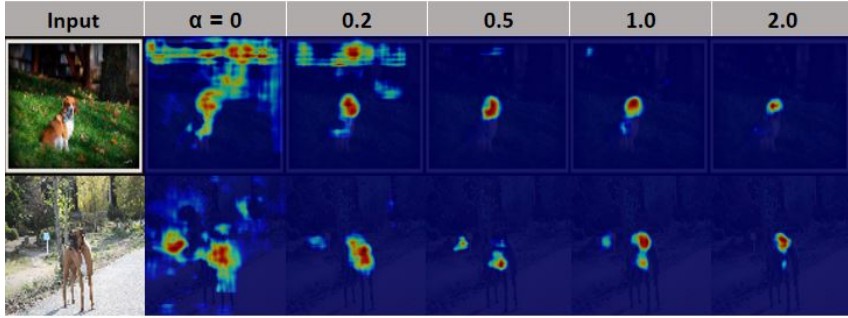

Figure 7: **Illustration of effect of the trade-off parameter ($\alpha$) in the Loss:** We see that, with increase in the value of $\alpha$, the model is able to focus more on the class-discriminatory causal feature in the input images.

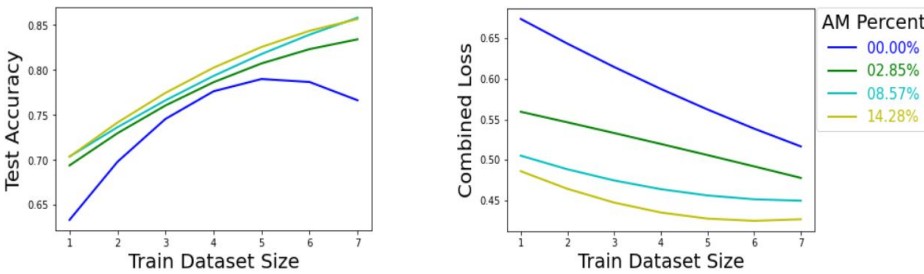

Figure 8: **Illustration of the effect of human guidance in the form of activation mask:** As we increase the amount of human guidance in the form of activation masks accuracy increases (left), and loss reduces at a greater rate with the increase in the training dataset size (in $x$-axis, 1 unit equals to 1000 samples). One observation is that, with additional guidance the models are able to learn efficiently with less data.

uncertainties in the model for learning causal features as the model is explicitly guided to focus on the features which are effective to discriminate.

To demonstrate this, we trained the CNN models with same architectures by varying the training dataset size in steps of 1000, from 1000 to 7000 training samples. In each case, we train models with 0, 200, 600 and 1000 activation masks as input. So basically we train 28 ($7\times4$) models in total. For a given dataset size we expect the models with more activation masks as input to have higher accuracy than those with less number of activation masks as input. We share the results in Fig. 8 in the form of two plots. To understand the average behavior of the models we present the curves which we regressed through the points representing the metric values for each model.

The first plot shows comparison of models on the basis of test accuracy. We see that the curve for the models with higher number of activation masks as input dominates over the curves of the models with less number of activation masks as input in almost all the train dataset size cases. This shows that the model is able to learn more with the same amount of data when it is provided with additional information as described in our approach.

The second plot considers the average of categorical cross entropy loss and the causally-focus loss as a metric for model comparison. This metric captures the correctness of the models both in terms of classification and causal feature extraction. We see that the models with more activation masks as input have lower loss than the models with less number of activation masks as input. We can also see that the lowest loss attained using 7000 training samples by the models with no activation masks as input is also attained by the model with just 200 activation masks as input using just 3000 to 4000 training samples. This confirms that, with more guidance the models are able to learn faster from less amount of data.

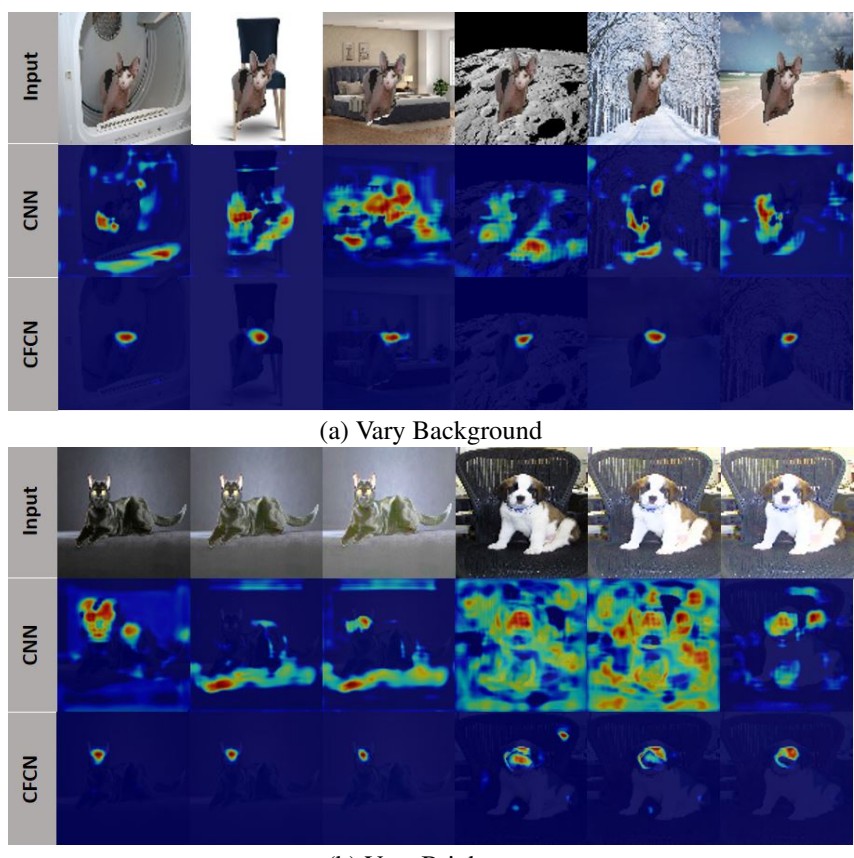

(a) Vary Background

(b) Vary Brightness

Figure 9: **Robust feature learning:** The above figures show the robustness of feature learning in CFCNs against changes in the background and the brightness of the input images.

### 4.7 RESULTS: ROBUSTNESS AGAINST ADVERSARIAL PERTURBATION OF IMAGES

Finally, we present another interesting experiment highlighting the robustness of the features learnt by the CFCN models in comparison to that of traditionally trained CNNs with the same architecture and identical experimental setup. We vary the background for the objects of interest and check how the two models perform in terms of visual results. We expect that in comparison to the traditional CNNs, the features learnt by CFCNs would be more stable across such image perturbations. As shown in Fig. 9(a), we see that the features learnt by CFCN are more robust to change in the context of the images. This further confirms the effectiveness of our method in learning causal features rather than relying of the context features for classification.

Similarly, we also conducted another experiment in this direction by varying the brightness of the input images. As shown in Fig. 9(b), we see that the features learnt by CFCNs are more robust in comparison to that of CNNs, for different brightness values of the input images.

## 5 CONCLUSION

In this paper we demonstrate that, through minimal human guidance it is possible to go beyond the traditional CNN architectures to avoid overestimating spurious correlations between labels and contextual features. Using such guidance in the form of activation masks, the proposed model CFCN is able to detect causal features in the images. This ability in turn helps CFCN not only to achieve higher accuracy with less data but also to be robust against several adversarial changes in the images.

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

# A    ACTIVATION MASK GENERATION

For manual activation mask generation, we have used the '*mpl_interactions*' library available in Python. As shown in Fig2, a user has to select the regions of interest in the given image which is then automatically converted into binary activation mask. These masks can be then stored along with the datasets for later use.

Further, in order to utilize the available annotations for activation mask generation, we can use the pixel level labels provided in the image segmentation datasets or the bounding box annotations provided in the object localization datasets. When using pixel level annotations, the pixel values representing the objects of interest can be set to 1s and rest other pixels can be set to 0s. Similarly, when using bounding box annotations, the pixels withing the bounding box regions need to be set to 1s while all other pixels outside the box regions need to be set to 0s. Few examples of such conversions are shown in Fig. 10.

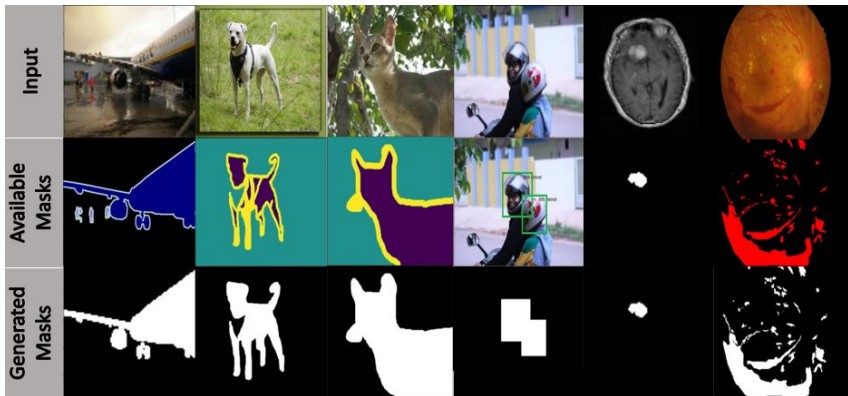

Figure 10: **Obtaining the activation masks:** The following table shows some examples of obtaining activation masks from different annotations available for some aligned tasks such as image segmentation and object localization. First row in the figure shows the input images, second row shows their corresponding available annotations and third row shows the activation masks generated from these annotations. As in the case of *Brain MRI* dataset, binary masks are readily available, which can be directly used for our purpose.

# B    ANALYSING THE LOSS FUNCTION

We present the Eq. 1 again for the purpose of analysis.

$$\mathcal{L} = \underbrace{-\sum_{i=1}^{C} y_i log(\hat{y}_i)}_{L_{cl}} + \underbrace{\alpha \left(1 - \frac{1}{F}\sum_{f=1}^{F}\left(\frac{\sum_{j=1}^{n_1}\sum_{k=1}^{n_2}(A \circ \hat{A}_f)_{j,k}}{\sum_{j=1}^{n_1}\sum_{k=1}^{n_2}(\hat{A}_f)_{j,k} + \epsilon}\right)\right)}_{L_{cf}}$$

The first term in the loss represents the standard categorical cross entropy loss ($L_{cl}$). The second term consists of the loss weight $\alpha$ and the causally-focus loss ($L_{cf}$). In this section we elaborate more on the second loss.

Intuitively, the causally-focus loss calculates the percentage of the activation values present in the output feature maps which are active in the context region of the images. More the activations present in the context regions, more will be the loss. In the optimal case, the model will have learnt only the causal features, in which case the loss will be equal to 0, which is the lowest. Contrarily, in the worst case, only the features outside the causal regions marked by the activation map will be active, making the loss equal to 1, which is the highest.

When using the CFCN for multi-label classification, the loss needs some modifications. An updated version of the loss for multi-label classification is given by Eq. 3. Basically, in the case of multi-label classification, each image will have '$C$' activation mask, one each corresponding to one of the $C$ classes in the network. Thus, it also requires the last convolutional layer to have number of filters to be equal to the number of classes in the dataset. This requirement is already satisfied in the case of the 'all convolutional nets'. Each filter corresponds to a class in the dataset. We calculate the causally-focus loss with respect to each feature map output and the corresponding activation mask for that class, as depicted in Eq. 3.

$$\mathcal{L} = -\sum_{i=1}^{C} y_i log(\hat{y}_i) \; + \; \alpha \; \left( 1 - \frac{1}{C} \sum_{f=1}^{C} \left( \frac{\sum_{j=1}^{n_1} \sum_{k=1}^{n_2} (A_f \circ \hat{A}_f)_{j,k}}{\sum_{j=1}^{n_1} \sum_{k=1}^{n_2} (\hat{A}_f)_{j,k} + \epsilon} \right) \right), \tag{3}$$

## C  DETAILED EXPERIMENTAL SETUP

Table 4: Detailed experimental setup for each dataset.

| Sr. | Dataset | Filters in conv. layers | DO | BS | LR | Epochs | $\alpha$ |
|-----|---------|-------------------------|-----|-----|------|--------|----------|
| 1. | *Oxford IIIT Pets* | 128,128,64,64,32,32,16,16 | 0.3 | 64 | 1e-4 | 500 | 1 |
| 2. | *Aeroplane-Cow* | 128,128,64,64,32 | 0.3 | 16 | 1e-4 | 200 | 1.5 |
| 3. | *Brain MRI* | 64,64,64,32,32,32 | - | 32 | 1e-5 | 150 | 3 |
| 4. | *IDRiD* | 32,32,16,16,8,8 | 0.3 | 5 | 1e-4 | 200 | 0.8 |

In this section we describe the detailed experimental setup used for training different models on each dataset. We have implemented all the models using Tensorflow v2.2. In all the experiments, we used custom CNN architectures for each dataset. Further, for each dataset, all the experiments were run by maintaining an identical experimental setup for all the models. Input images from all the models except the *IDRiD* dataset were resized to the shape 96×96 while those belonging to the *IDRiD* dataset were resized to the shape 250×175. The number of filters used in each convolutional layers of these architectures is given in Tab. 4. Every convolutional layer is followed by a batch normalization layer and later by a dropout layer (except for the *Brain MRI* dataset). We use a dropout(DO) value of 0.3. The last dropout layer is followed by a convolutional layer with 1×1 filter size and 16 filters in the case of CFCN-F models and '$C$' filters in the case of CFCN-C, where '$C$' is the number of classes in the given dataset. As $IDRiD$ is a multi-label classification dataset, the $1 times 1$ convolutional layer in this case has 3 filters, for both CFCN-F and CFCN-C. The 1×1 convolutional layer is then followed by a flatten layer in the case of CFCN-F, while for the CFCN-C models, we use global average pooling. In the case of CFCN-F the flatten layer is followed by the output layer which is a fully connected layer with $C$ neurons in it. In CFCN-C, global average pooling layer acts as the output layer. We have used 'relu' activation in all the intermediate layers. For output layer, we use 'softmax' activation in the case of multi-class classification and 'sigmoid' in the case of multi-label classification. Details about the batch size (BS), learning rate (LR), number of epochs and the value for loss weight parameter $\alpha$ introduced in Eq. 1 and 2, for each dataset are given in Tab. 4.

For the *Oxford IIIT Pets* dataset and the *IDRiD* dataset, we do not down-sample the input images throughout the network. For the *Aeroplane-Cow* and *Brain MRI* datasets, we down-sample the images by using a stride of 2 in every second and every third convolutional layer in the respective dataset case. While calculating the causally focus loss, we can resize the input activation masks from input size to the size of the output feature maps. Conversely, we can also resize the output feature maps to the input size. In case of the Oxford IIIT Pets and IDRiD, the input activation masks and the output feature maps have the same size as the input (ie. $(m_1 \times m_2) = (n_1 \times n_2)$), so no resize operation is required. For the *Aeroplane-Cow* dataset, we resize the output activation mask to the input size while for *Brain MRI* dataset, we resize the input activation masks to the size of output feature maps.

# D  ADDITIONAL RESULTS

## D.1  BENEFITS IN VISUAL EXPLANATION

In this subsection, we present the benefits of using the all convolutional nets. As discusses earlier, the all convolutional nets are known for their architectural simplicity and their ability to retain the spatial structure of the images throughout the network. Additionally, the last convolutional layer of these nets generate feature maps, each of which corresponds to one particular class in the dataset. Thus, each feature map retains the features in the input images which are similar to those of the images belonging to the class represented by that feature map. This fact is clearly highlighted in Fig. 11. We see that, for the first image of digit '0', the most similar digits are the digit '2' and digit '6'. Similarly, for the second image of input digit '9', the closest digits are '1', '4' and '7'.

Due, to this explanations similar to the Grad-Cam are obtained in just the single forward pass through the network, thereby eliminating the need of back-propagating through the network for gradient calculations. When using CFCN with the all convolutional nets, the additional benefit is that, each feature map output highlights only the *class-specific causal features* present in the images, thereby enhancing the explainability of these models.

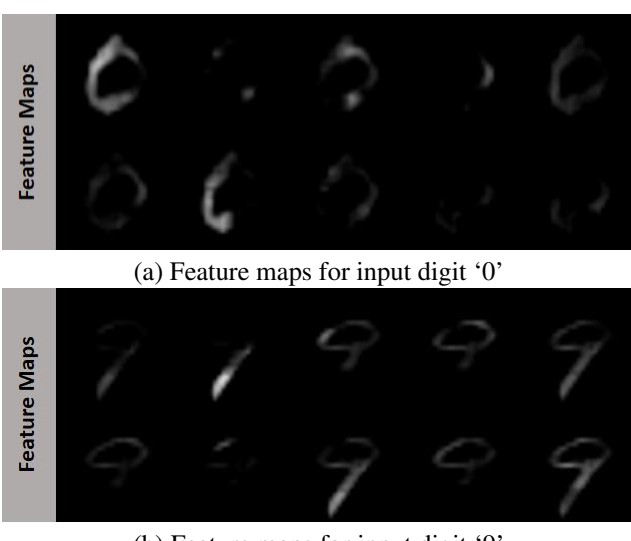

(a) Feature maps for input digit '0'

(b) Feature maps for input digit '9'

Figure 11: **Enhanced visual explanations:** This figure demonstrates the benefits of all convolutional nets in terms of visual explanations. The feature map outputs for the input digits (a) 6 and (b) 9 are given. The images represent the classes 0 to 9 in row major order from left to right.

## D.2 Additional Training Plots

In Fig. 12, we present the training plots for the remaining two dataset: *Aeroplane-Cow* and *IDRiD*. These plots also confirm empirically that the modified loss in CFCN attains convergence after a specified number of epochs.

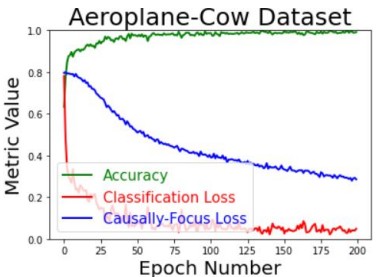 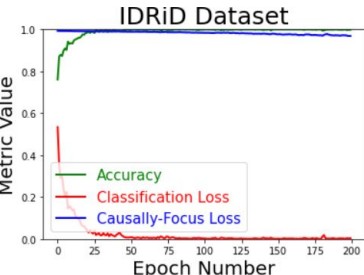

Figure 12: **Convergence:** Causally-focus loss does not affect the convergence in training CNNs. Demonstration using the training plots for *Aeroplane-Cow* and *IDRiD* datasets.

## D.3 Comparison the Causally-Focus Loss on the Benchmarks

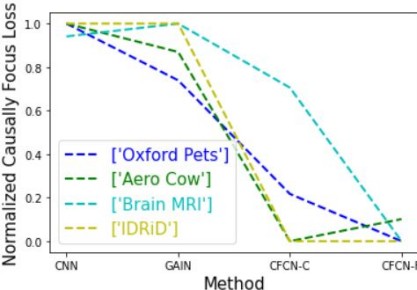

Figure 13: **Comparing the causally-focus loss:** In the above plot, we compare the values of the causally-focus loss for different models on the benchmark datasets. We observe that, our models perform better than baselines in all the dataset cases.

In Fig. 13, we present the comparison of the causally-focus loss for different models on different datasets. The $x$-axis represents the models while the $y$-axis represents the loss values. For better visualization, we have performed min-max normalization over the causally-focus losses of the models for a given dataset. We observe that the causally-focus loss is the lowest mostly in the case of CFCN-F models. Also, the CFCN-C models perform more or less similar to these models. This further confirms that our method is successful in eliminating the spurious correlations based on the context features, thereby providing reliable predictions.

### D.4 CAUSAL FEATURE LEARNING IN CFCN

In Fig. 14, we present few additional results comparing the heatmaps generated by the traditionally trained CNN and the CFCN on the *Bike-Helmets* dataset. This dataset consists of about 764 images in total, of which we have used 500 samples for testing.

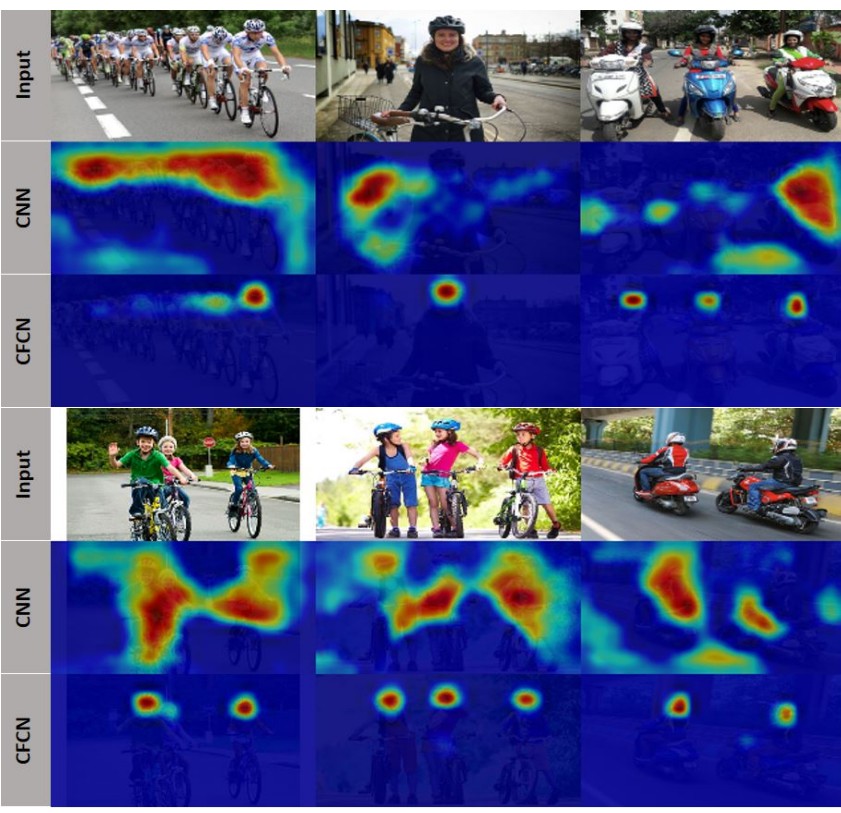

Figure 14: Additional qualitative results presenting the Grad-Cam heatmaps generated for the *Bike-Helmets* dataset provided by Make-ML.

## D.5 VARY BACKGROUND

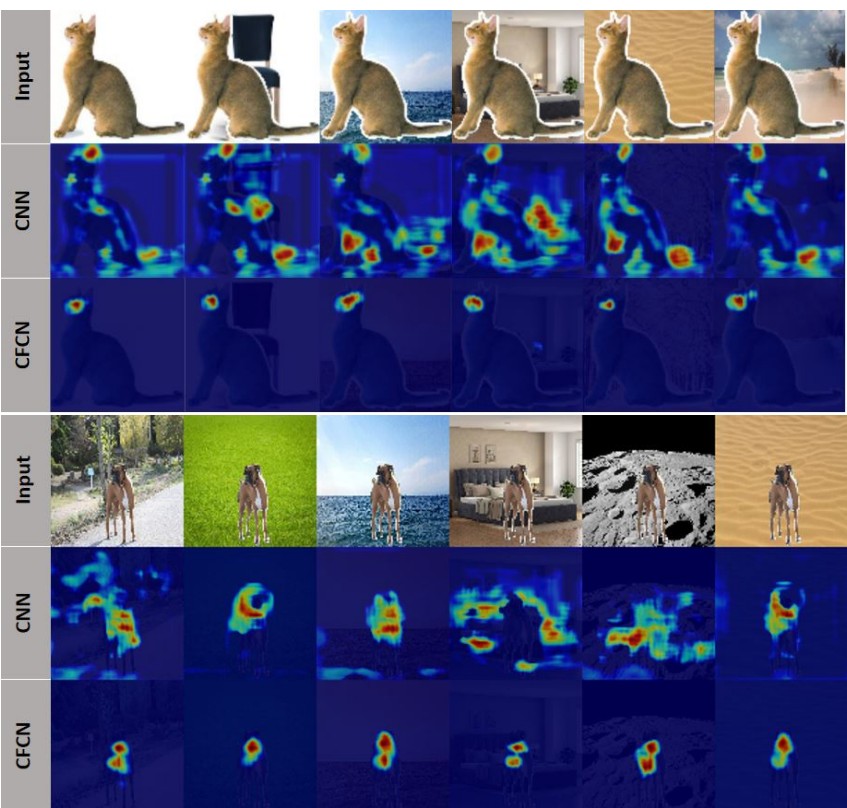

Figure 15: Additional visual results comparing the robustness of features learnt by CNN and CFCN for the vary-background experiment. The results show that CFCN consistently outperforms CNN in terms of robust feature learning.

In Fig. 15, we present few more results comparing the traditionally trained CNNs and the CFCN models for the vary-background experiment. We observe that CFCN are able to extract the class discriminatory causal features in the images in all the cases and it is least affected by the changes in the context regions of the input images.

