# OpenReview forum: "Causally Focused Convolutional Networks Through Minimal Human Guidance"
_ICLR.cc/2022/Conference — ICLR 2022 Submitted_

### Official Review · Reviewer_nnUS · 2021-11-01

**Correctness:** 3
**Technical Novelty And Significance:** 2
**Empirical Novelty And Significance:** 2
**Recommendation:** 3
**Confidence:** 3

**Main Review:**

**General comments:**

The masking approach is very similar to binary segmentation (e.g. foreground vs background). How well the model would work if the current "masking loss" would be substituted by a binary segmentation loss? Such approaches have been extensively studied in the literature, see e.g. https://arxiv.org/pdf/2004.07054.pdf

It is unclear why the authors decided to use grad-cam given a large body of explainability work (see e.g. https://medium.com/analytics-vidhya/cnns-explainability-papers-review-5ed380577c64)?

The limitations of Grad-Cam are not discussed either. Is Grad-Cam the best tool to detect causal features?

Would the approach work with multiple objects?

The limitations to the model are not discussed. Adding discussion of limitations would make the paper stronger.

Would similar approach work with non-CNN based models, e.g. ViT?


**Specific comments:**

_Introduction:_

"... these systems are expected to be justifiable..." Could the authors clarify this statement? Do the authors mean the decision to be explainable or interpretable? Is it always the case that this expectation is in place for all scenarios discussed in this paragraph?

"... and often extract features greedily following that principle.." What do the authors mean by greedily in this context? It is also a bit unclear to which principle the authors are referring to.

"In order to eliminate the effect of non causal correlations, CNNs need to be trained on huge datasets..." Could the authors provide a citation for this statement? Can we always remove non causal correlations just by scaling up the dataset?

"... minimal human guidance." Is providing a binary mask really a minimal human guidance? Would it be easier to just provide yes no answers, e.g. is this gradCAM visualization respecting the causal relations for his object?


_Methodology:_

Fig. 3: It is a bit unclear at which point the binary mask are applied in the CNN pipeline. The figure could be improved by depicting explicitly at which point the multiplication of features with binary mask is performed.

From the current description, it is unclear how the model works at inference time. Do we still need binary masks at inference time?

Eq. 1 lacks explanation. In particular, why this particular loss is beneficial for the task? Could we design the loss differently? What makes this particular formulation important?

_Experiments:_

It would be nice to see the effect of how exact the mask has to be in order to notice the described effect, eg. how precisely the object has to be segmented to provide meaningful causal features?

Fig. 4: It is interesting to notice that the highlighted features do not capture the whole plane, only some parts of the plane seems to be causally relevant. Adding a discussion on why some pixels are more relevant than others would be beneficial.

Tab 3. There is no measure of std or variance.

Fig 6. The convergence claim (the modified loss does not affect convergence) is not supported by the plot. To support the claim, the convergence plot of vanilla CNN should be depicted. It would also be nice to state which dataset split has been used to obtain them.

Fig 8. It is unclear what AM stands for. This figure also indicates that the ablation was run on the test set, it would be more principled to run all ablations on the validation set.


**Summary Of The Paper:**

The paper proposes to alternate the loss of classification CNNs by adding additional loss term that focuses the model's attention on the object present in the image rather than on the background. To do so, the model is provided with an additional input (a binary mask) that is used to guide the learning of the model. The approach is validated on four datasets showing some benefits in terms of classification performance on small-scale datasets while decreasing the classification performance on larger scale datasets.

**Summary Of The Review:**

The paper tackles an interesting problem of making the model more robust to spurios correlations. Although the reviewer finds the problem interesting and relevant for the ICLR community, the reviewer thinks that the paper in its current state is below the acceptance threshold.
The methodological contribution is rather minor. The approach requires collections of binary masks that might not be available for all datasets. It is also a bit unclear what the community learns from this paper -- it is expected to have more object centered gradCAM activations by using object binary masks. Moreover, the presentation of the paper could be significantly improved. For details see main review.

---

> ### Author Response · Authors · 2021-11-19
> **Authors response to reviewer nnUS [Part 1/2]**
>
> We are thankful for the valuable feedback. We clarify the doubts below.
>
> >**Q1.** How well the model will perform if we replace the causally-focus loss with binary segmentation loss [1]?
>
> **Ans:** We have checked with binary segmentation loss but found that the model is not able to detect causal features effectively while achieving good classification accuracy.
>
> >**Q2.** Is Grad-Cam the best tool to detect causal features? The limitations of Grad-Cam are not discussed either.
>
> **Ans:** We would like to clarify that Grad-Cam is not a tool to detect causal features.  We do not use Grad-Cam in our method for learning causal features.  We just use it to obtain the heatmaps for output comparison. Learning of causal features happens because of the modified loss function given in Equation 1 and 2.
>
> >**Q3.** "... these systems are expected to be justifiable..." Could the authors clarify this statement?
>
> **Ans:** We mean that these systems need to justify their decisions with human acceptable explanations. This is important in risk critical areas as multiple things may be at stake of a single decision.
>
> >**Q4.**  "... and often extract features greedily following that principle.." What do the authors mean by greedily in this context? It is also a bit unclear to which principle the authors are referring to.
>
> **Ans:** By greedily, we mean that the CNN model learns those features from the images which are most class discriminatory and which contribute more towards reducing the categorical cross-entropy loss. This encourages correlational learning in CNNs. The principle that we speak of is  as follows : CNNs used for image classification are usually trained to optimize just the categorical cross-entropy loss, which compresses the data while preserving the mutual information present in the form of any regularity in the data [2,3,4,5].
>
> >**Q5.** "In order to eliminate the effect of noncausal correlations, CNNs need to be trained on huge datasets..." Could the authors provide a citation for this statement? Can we always remove noncausal correlations just by scaling up the dataset?
>
> **Ans:** Mere scaling won't help. By scaling, we mean that the dataset includes more images bringing it close to the true class distribution of the various image classes in the dataset. Figure 8 shows that as we increase the dataset size, the combined loss is decreasing.
>
> >**Q6.** Would it be easier to just provide yes no answers, e.g. is this Grad-Cam visualization respecting the causal relations for his object?
>
> **Ans:** The Yes/No type of guidance would require human intervention during online training which might significantly slow down the training procedure. Also, Yes/No kind of guidance cannot help to point towards the causal features in the images. This raises questions about its applicability.
>
> >**Q7.** Fig. 3: It is a bit unclear at which point the binary masks are applied in the CNN pipeline.
>
> **Ans:** We never apply masks to the images or the feature map outputs. Instead, we propose to use these masks to quantify the percentage of activations that are present in the context regions of the image as calculated by the causally-focus loss in Equation 1 and 2.
>
> >**Q8.** From the current description, it is unclear how the model works at inference time. Do we still need binary masks at inference time?
>
> **Ans:** No, we do not use activation masks during inference (refer solid lines in Figure 3). Due to the modified loss, the model parameters are updated in such a way that it becomes capable of using just the causal features for classification.
>
> >**Q9.** Equation 1 lacks explanation. In particular, why is this particular loss beneficial for the task? Why not some another loss?
>
> **Ans:** The main objective of the causally-focus loss is to quantify the percentage of activations in the feature map of the last convolutional layer that lie in the background regions of the image. We have also explored binary segmentation loss and dice loss but they did not help much. The intuition behind the causally focus loss is straightforward i.e. punish if regions outside guided regions are given higher weightage.

---

> ### Author Response · Authors · 2021-11-19
> **Authors response to reviewer nnUS [Part 2/2]**
>
>
>
> >**Q10.** How precise the masks need to be?
>
> **Ans:**  The masks need not be very much precise. It is sufficient to cover distinguishing parts within the causal regions.
>
> >**Q11.** Fig. 4: Why the highlighted features do not capture the whole plane?
>
> **Ans:** The model optimizes causally focus loss along with the categorical cross entropy loss. The 1st loss finds the causal region, and then then 2nd part finds the distinguishing parts inside the causal regions.
>
> >**Q12.** Fig 6. To support the claim, the convergence plot of vanilla CNN should be depicted. Which dataset split was used?
>
> **Ans:** We have cross checked these results, and as expected CNN loss converges steadily. We did not add them due to space constraints.
>
> >**Q13.** Fig 8. It is unclear what AM stands for. Ablation on validation data would be more meaningful.
>
> **Ans:** AM stands for activation mask count. Plots for validation data also show a similar trend.
>
>
> >**Q14.** Would the approach work with multiple objects?
>
> **Ans:** Yes, the method will also work in the multi-object case with modifications as defined in Section B of the appendix of the paper.
>
> **REFERENCES**
>
> [1] Wu, Yu-Huan, et al. "Jcs: An explainable covid-19 diagnosis system by joint classification and segmentation." IEEE Transactions on Image Processing 30 (2021): 3113-3126.
>
> [2] Naftali Tishby and Noga Zaslavsky. “Deep learning and the information bottleneck principle.” In: 2015 IEEE Information Theory Workshop (ITW). IEEE. 2015, pp. 1–5.
>
> [3] Ravid Shwartz-Ziv and Naftali Tishby. “Opening the black box of deep neural networks via information.” In: arXiv preprint arXiv:1703.00810 (2017).
>
> [4] Gregory Chaitin. On the intelligibility of the universe and the notions of simplicity, complexity, and irreducibility. Akademie Verlag, 2015.
>
> [5] Léonard Blier and Yann Ollivier. “The Description Length of Deep Learning models.” In: Advances in Neural Information Processing Systems. Ed. by S. Bengio, H. Wallach, H. Larochelle, K. Grauman, N. Cesa-Bianchi, and R. Garnett. Vol. 31. Curran Associates, Inc., 2018. url: https://proceedings.neurips.cc/paper/2018/file/3b712de48137572f3849aabd5666a4e3-Paper.pdf

---

### Official Review · Reviewer_teVu · 2021-11-02

**Correctness:** 2
**Technical Novelty And Significance:** 2
**Empirical Novelty And Significance:** 2
**Recommendation:** 3
**Confidence:** 4

**Main Review:**

In this work, the authors design a causally focused CNN framework through human guidance. By utilizing the auxiliary guidance, the proposed method can particularly focus on the information useful for the classification tasks. Extensive experiments on several datasets have indicated the effectiveness of the proposed method.

Strength:
+ The proposed method can achieve appealing performance compared with normal classification framework without guidance.
+ Several feature visualization results have been presented to indicate the effective of the proposed method.

Weakness:
- There lack discussions on the potential guidance which can be used to improve the model performance on learning causal features. It seems that only segmentation masks are introduced in the manuscript.
- The potential effect of the proposed method is questionable. As introduced in Section 4.3, the coarse mask boundaries are obtained by a simple regression algorithm. However, the region of interest (ROIs) in the images are sometimes hard to segment. For example, segmenting Multiple Sclerosis lesions in brain MRI is challenging, even a well-trained CNN framework cannot always produce accurate segmentation masks. Under this situation, can the proposed framework still work?
- The overall methodology design lacks novelty. The causally-focus loss in Equation 1 is similar to the Dice loss, which is widely used for medical image segmentation, such as:
F. Milletari, N. Navab, S.A. Ahmadi, “V-Net: Fully Convolutional Neural Networks for Volumetric Medical Image Segmentation”, 3DV2016, pp565-571, 2016.
- For each dataset, there lacks an introduction on the specific classification task, and the data split details.
- Introducing an auxiliary segmentation task learning might induce computational cost. Given the unstable performance gain in Table 3, it is questionable whether the proposed framework is worthwhile to use in real practice.

Questions:
(1) What if replacing the causally-focus loss in Equation 1 with the normal cross-entropy loss and dice loss?
(2) Compared with the model without the auxiliary segmentation task learning, how much auxiliary computational cost will the proposed method bring?


**Summary Of The Paper:**

This work aims at improving the classification tasks via segmentation learning. The methodology and experiments are problematic.

**Summary Of The Review:**

As discussed in the review comments, I think this manuscript lacks technique novelty, has potential limits to more challenging tasks, and has less practical usages. In addition, the overall manuscript is not clearly written. Therefore, I recommend rejecting the paper.

---

> ### Author Response · Authors · 2021-11-19
> **Authors response to reviewer teVu**
>
> Thank you for the valuable comments. We clarify the doubts below.
>
> >**Q1.** “This work aims at improving the classification tasks via segmentation learning...”
>
> **Ans:** We would like to clarify that this is not the objective of the work. We do not learn segmentation, or we apply segmentation. We provide humans guidance through activations masks to learn causal features. These masks indicate the causal region of images.
>
> >**Q2.** Discussion on potential guidance for creating activation masks is missing.
>
> **Ans:** We would like to clarify that, creation of activation masks is discussed in Section 3.2 and Figures 2 and 10.
>
> >**Q3.** The potential effect of the proposed method is questionable. As introduced in Section 4.3, the coarse mask boundaries are obtained by a simple regression algorithm. However, the region of interest (ROIs) in the images is sometimes hard to segment.
>
> **Ans:** We believe that there is some misunderstanding in this case. We do not perform regression while manually generating the activation masks (refer Section 3.2) but obtain them from human experts. By the regress in 4.3 we meant the curves in the results (Fig 8).
>
>
> >**Q4.** What if we replace the causally-focus loss in Equation 1 with the Dice loss?
>
> **Ans:** In our preliminary experiments, we have experimented with the Dice loss as well as the binary segmentation loss [4], but both these losses are inclined towards segmentation and we have observed that the model fails in the classification task.
>
>
> >**Q5.** Introducing an auxiliary segmentation task learning might induce computational cost.
>
> **Ans:** We believe that there is a misunderstanding here. We are not performing segmentation at all in our method. We have modified the training loss of the CNNs to incorporate additional guidance in the form of activation masks. Our mere objective is to reliably learn causal features for the task of image classification.

---

> > ### Comment · Reviewer_teVu · 2021-11-29
> > **Reply to the authors' rebuttal**
> >
> > Thanks to the authors for their response. However, I think the authors did not address my concerns.
> >
> > First, I do not think my review comments contain any misunderstandings. Based on the definition of the activation masks, they are just the ground truth for semantic segmentation. In addition, the L_{cf} in Equations 1 and 2 has a similar look to the Dice loss, i.e, calculating the intersection between the ground truth and the predictions on the numerator. To this end, although the authors did not say they were doing anything related to the segmentation directly in the paper, the objective of the L_{cf} is similar to learning segmentation using the coarsely annotated masks.
> >
> > Second, based on the definition of Figure 10, it seems that the proposed method would utilize the auxiliary annotations on segmentation masks or detection bounding boxes. To this end, I think the practical usages of the proposed method are quite limited. Normally, obtaining the pixel-level and object-level annotations for the images incurs a high cost.
> >
> > Given the inconvincible feedback and the remaining concerns regarding this work, I will lower my original rating.

---

### Official Review · Reviewer_HtjX · 2021-11-02

**Correctness:** 3
**Technical Novelty And Significance:** 2
**Empirical Novelty And Significance:** 2
**Recommendation:** 5
**Confidence:** 4

**Main Review:**

Strength:
1> The proposed method is very simple to understand and adapt.

2> A small percent of images with binary mask labels can be used to make this work as shown in experiments.

3> The proposed method can be beneficial for application areas such as medical imaging where “why” matters equally as “what”.

Weakness:

1> Method requires some user intervention which might be possible for simple applications as natural images but not possible without expertise in some applications such as medical.

2> Sometimes relative features are also important such as the correlation between background and foreground. For example, a foreground with a completely irrelevant background does not make sense (a cat flying in outer space).

3> Authors have experimented with near-to-good masks but in reality, masks could be noisy (partially or completely wrong). No study is done.

4> The proposed method is closely related to explainable artificial intelligence (XAI) methods but there is no comparison with XAI method.

5> Authors claim for the proposed method to work better on the small dataset: A small dataset with less number of classes is not very difficult for a model to learn. The model difficulty associated with dataset difficulty such as a high number of classes with a small number of images in each class, intra-class variations, inter-class similarity, etc. are factors of dataset difficulty, not just small dataset. I believe the degraded performance on the Oxford pet dataset is because of dataset difficulty with a large number of classes and similarity between different classes.

Comments and suggestions:

1> Authors Could try with soft masks instead of hard binary masks for background relation factor and compare the results.

2> Also some noisy masks ablation studies will be interesting to see.

3> It is not clear from the paper that if all the classes (at least a few images per class) in the dataset have binary masks for training or some classes have zeros masked training. It will be good to see if some classes are left with zero masks, how they perform during testing and how they adapt knowledge from other-classes mask training.

3> Intrinsic XAI models can be good candidates to check model performance and activation/heatmaps for better comparison.

4> Authors should try the proposed method on a difficult dataset such as any fine-grained dataset with a large number of classes (e.g., Stanford-cats, cub-200). However, this is rare in medical imaging which authors claim to be the main objective of the proposed method. Still, it would be better to see the method’s low points (if any in this regard) for clear future use.

5> It is interesting that how model focus (heatmap) are reduced in the smallest area (which might be responsible for a particular class) irrespective of providing full foreground as binary mask (fig 4 and 5). Any comments on this?

6> How is proposed method is different from the extension version of GAIN [1] (pixel wise and bbox masks) is not explained anywhere.

[1] Li et al. Guided attention inference network. IEEE TPAMI, 42(12):2996–3010, 2019.


**Summary Of The Paper:**

This paper proposes a user-guided training for CNN to focus more on casual features which are based on human perspective and termed as casually focused convolutional neural network. Human guidance is used to make a rough binary segmentation mask of foreground objects/pixels.

For an input image, mask and label are both utilized to compute final loss. Final loss is composed of 1) activation loss which is computed using the binary mask and last convolution layer features and 2) cross-entropy loss which is calculated using predicted probability and class label of the input image. Images for which binary mask is not available, dummy mask is created as all foreground pixel. Two types of networks, all convolutional layers (CFCN-C) and fully-connected classifiers (CFCN-F) are utilized with the proposed methodology.

Experiments are performed on a total of 4 datasets of natural and medical images. Accuracy and visual activations are compared with baseline CNN, GAIN [1], and both CFCN-C and CFCN-F.


**Summary Of The Review:**

1> Difference between GAIN and the proposed method is not clear. What are the technical improvement over GAIN and how are they working?

2> Experimental section is weak and could be improved as per suggestions. Also, there is no comparison with intrinsic XAI methods. Proper evaluation can make things more clear.

---

> ### Author Response · Authors · 2021-11-19
> **Authors response to reviewer HtjX**
>
> We are thankful for your valuable feedback, and clarify the doubts below.
>
> >**Q1.** The drawback of the proposed method is that it needs intervention from a human expert.
>
> **Ans:**
> The objective of the work is to enable CNN to detect features that are causal from the human perspective. It has been observed that traditional ways of training CNN cannot address the given objective. Now to extract features that humans find causal can be done only through some guidance from humans. Hence, human intervention is not a drawback here but a feature or attribute of the method. Moreover, the exciting thing is that even providing guidance for only a small number of cases is sufficient (Fig 8).
>
> >**Q2.** Sometimes background and foreground features together make a valid image. (Eg. a cat flying in outer space may not be valid).
>
> **Ans:**
> We would like to clarify that, we focus only on labeling images through detecting causal features. Checking the validity of images is a separate task and is out of the scope of this work. However, CNN is equally incapable in this aspect.
>
> >**Q3.** Some noisy mask ablation studies will be interesting to see.
>
> **Ans:** In CFCN, we use the activation masks generated by a human expert. These masks can be slightly noisy (refer Figures 2, 10). Additionally, masks obtained from bounding box annotations can be comparatively noisier (refer column 4 of Figure 10). But as we expect the annotations to come from human experts, we can assume that, even though they might not be perfect, they will at least contain the class discriminatory causal features in them. The results in Figures 7 and 9 show that CFCNs don't rely on all the features from the causal regions mentioned by the human expert but only pick the most class discriminatory causal features from the images. Refer Figure 14 which shows that CFCNs are able to capture the presence of helmets as important features even when we used coarse masks generated from bounding boxes. Also, the noisy mask ablation study is similar to the problem of learning from noisy labels and we can explore it as a part of our future work.
>
> >**Q4.** Intrinsic XAI models can be good candidates to check model performance and activation/heatmaps for better comparison.
>
> **Ans:** During our literature survey we hardly came across any existing method that proposed to reliably learn the causal features from the images. Actually, learning causal features is our main objective which in turn provides explainability.
>
> >**Q5.** Try challenging small datasets with more classes and fewer samples per class(e.g., Stanford-cats, cub-200).
>
> **Ans:** We have explored the performance by reducing the number of samples, and observed a positive outcome. However, it will be interesting to explore by increasing the number of classes.
>
> >**Q6.** Can we use soft masks instead of hard binary masks?
>
> **Ans:**
> It is not very clear what the reviewer meant by the soft mask. We guess it is an activation mask with values between 0 and 1 instead of 0 or 1. However, this will not make much difference, as it will still provide guidance. However, it will be more interesting if we associate probability or confidence in terms of the soft mask. But we have not explored that. Thanks for the suggestion.
>
> >**Q7.** How will the model perform if we do not provide masks for a few classes?
>
> **Ans:** We have observed that CFCN can learn from activation masks of a very small number of samples by propagating this information across samples in a given class. However, we have not observed the learning to be benefited from activation masks of other classes, and this is also aligned with our hypothesis.
>
> >**Q8.** Why is the model focusing on small regions even when masks for entire objects of interest are provided?
>
> **Ans:** This is an interesting question. The model still tries to optimize to learn distinguishing features. However, because of the causally focus loss, it tries to find the distinguishing features within the valid region.
>
> >**Q9.** How does the proposed method differ from the extended versions of GAIN [1]?
>
> **Ans:** Here are a few key differences between CFCNs and the extended versions of GAIN:
>
> _Objective_: GAIN is proposed for the task of weakly supervised semantic segmentation while CFCNs aim at reliably learning causal features for classification.
>
> _Architecture_: the GAIN architecture is complex with 3 parallel streams of parameter sharing networks [1], while CFCN requires only slight modifications to the underlying CNN architecture.
>
> _Computation_: GAIN is computationally expensive as it proposes to make Class Activation Maps end to end trainable during model training. CFCNs do not require such heavy computation.
>
> **REFERENCES**
>
> [1] Li, Kunpeng, et al. "Tell me where to look: Guided attention inference network." Proceedings of the IEEE Conference on Computer Vision and Pattern Recognition. 2018.

---

> > ### Comment · Reviewer_HtjX · 2021-11-24
> > **Reply to Authors**
> >
> > I thank authors for replying most comments. However, authors have mentioned most reviewer's comments directly but it should be implemented in order to prove more reliability of the work.
> > Q3. From noisy masks, I meant wrong masks which partially or entirely cover different region. The mask provided by authors contain full region of interest along with some background.
> > Q4. Authors can look for similar papers such as: Li, Liangzhi, et al. "SCOUTER: Slot attention-based classifier for explainable image recognition." ICCV 2021.
> >
> > For future, I would suggest authors to implement Q3, Q4, Q5, Q7 in paper and a better explanation of Q8.

---

### Decision · Program_Chairs · 2022-01-20

**Decision:**

Reject

**Comment:**

Although this paper is on an interesting topic, there is a consensus that this paper is below the bar for acceptance. My advice is to take take criticisms of the reviewers seriously, add the extra experiments, rewrite the paper and then submit it to a different conference. If the authors feel that the reviewers misunderstood their paper, please remember that the level to which they were able to understand it is also a function of how the paper is written.